# DIFFUSION PROBABILISTIC FIELDS

**Peiye Zhuang**[1†]**, Samira Abnar**[2] **, Jiatao Gu**[2]**, Alexander G. Schwing**[3]**,
Joshua M. Susskind**[2] **, Miguel Ángel Bautista**[2]
[1]Stanford University     [2]Apple     [3]University of Illinois at Urbana-Champaign
[1]peiye@stanford.edu [2]{abnar, jgu32, jsusskind,
mbautistamartin}@apple.com [3]aschwing@illinois.edu

## ABSTRACT

Diffusion probabilistic models have quickly become a major approach for generative modeling of images, 3D geometry, video and other domains. However, to adapt diffusion generative modeling to these domains the denoising network needs to be carefully designed for each domain independently, oftentimes under the assumption that data lives in a Euclidean grid. In this paper we introduce Diffusion Probabilistic Fields (DPF), a diffusion model that can learn distributions over continuous functions defined over metric spaces, commonly known as *fields*. We extend the formulation of diffusion probabilistic models to deal with this field parametrization in an explicit way, enabling us to define an end-to-end learning algorithm that side-steps the requirement of representing fields with latent vectors as in previous approaches (Dupont et al., 2022a; Du et al., 2021). We empirically show that, while using the same denoising network, DPF effectively deals with different modalities like 2D images and 3D geometry, in addition to modeling distributions over fields defined on non-Euclidean metric spaces.

## 1 INTRODUCTION

Diffusion probabilistic modeling has quickly become a central approach for learning data distributions, obtaining impressive empirical results across multiple domains like images (Nichol & Dhariwal, 2021), videos (Ho et al., 2022) or even 3D geometry (Luo & Hu, 2021). In particular, Denoising Diffusion Probabilistic Models (often referred to as DDPMs or diffusion generative models) (Ho et al., 2020; Nichol & Dhariwal, 2021) and their continuous-time extension (Song et al., 2021b) both present a training objective that is more stable than precursors like generative adversarial nets (GANs) (Goodfellow et al., 2014) or energy-based models (EBMs) (Du et al., 2020). In addition, diffusion generative models have shown to empirically outperform GANs in the image domain (Dhariwal & Nichol, 2021) and to suffer less from mode-seeking pathologies during training (Kodali et al., 2017).

A diffusion generative model consists of three main components: the forward (or *diffusion*) process, the backward (or *inference*) process, and the denoising network (also referred to as the *score network*[1] due to its equivalence with denoising score-matching approaches Dickstein et al. (2015)). A substantial body of work has addressed different definitions of the forward and backward processes (Rissanen et al., 2022; Bansal et al., 2022; Song et al., 2021a), focusing on the image domain. However, there are two caveats with current diffusion models that remain open. The first one is that data is typically assumed to live on a discrete Euclidean grid (exceptions include work on molecules (Hoogeboom et al., 2022) and point clouds (Luo & Hu, 2021)). The second one is that the denoising network is heavily tuned for each specific data domain, with different network architectures used for images (Nichol & Dhariwal, 2021), video (Ho et al., 2022), or geometry (Luo & Hu, 2021).

In order to extend the success of diffusion generative models to the large number of diverse areas in science and engineering, a unification of the score formulation is required. Importantly, a unification enables use of the same score network across different data domains exhibiting different geometric structure without requiring data to live in or to be projected into a discrete Euclidean grid.

---

†Work was completed while P.Z. was an intern with Apple.
[1]We use the terms score network/function and denoising network/function exchangeably in the paper.

To achieve this, in this paper, we introduce the Diffusion Probabilistic Field (DPF). DPFs make progress towards the ambitious goal of unifying diffusion generative modeling across domains by learning distributions over continuous functions. For this, we take a functional view of the data, interpreting a data point $x \in \mathbb{R}^d$ as a function $f : \mathcal{M} \to \mathcal{Y}$ (Dupont et al., 2022a; Du et al., 2021). The function $f$ maps elements from a metric space $\mathcal{M}$ to a signal space $\mathcal{Y}$. This functional view of the data is commonly referred to as a *field* representation (Xie et al., 2022), which we use to refer to functions of this type. An illustration of this field interpretation is provided in Fig. 1. Using the image domain as an illustrative example we can see that one can either interpret images as multidimensional array $x_i \in \mathbb{R}^{h \times w} \times \mathbb{R}^3$ or as field $f : \mathbb{R}^2 \to \mathbb{R}^3$ that maps 2D pixel coordinates to RGB values. This field view enables a unification of seemingly different data domains under the same parametrization. For instance, 3D geometry data is represented via $f : \mathbb{R}^3 \to \mathbb{R}$, and spherical signals become fields $f : \mathbb{S}^2 \to \mathbb{R}^d$.

In an effort to unify generative modeling across different data domains, field data representations have shown promise in three recent approaches: From data to functa (Functa) (Dupont et al., 2022a), GEnerative Manifold learning (GEM) (Du et al., 2021) and Generative Adversarial Stochastic Process (GASP) (Dupont et al., 2022b). The first two approaches adopt a latent field parametrization (Park et al., 2019), where a field network is parametrized via a Hypernetwork (Ha et al., 2017) that takes as input a trainable latent vector. During training, a latent vector for each field is optimized in an initial reconstruction stage (Park et al., 2019). In Functa (Dupont et al., 2022a) the authors then propose to learn the distribution of optimized latents in an independent second training stage, similar to the approach by Rombach et al. (2022); Vahdat et al. (2021). Du et al. (2021) define additional latent neighborhood regularizers during the reconstruction stage. Sampling is then performed in a non-parametric way: one chooses a random latent vector from the set of optimized latents and projects it into its local neighborhood before adding Gaussian noise. See Fig. 2 for a visual summary of the differences between Functa (Dupont et al., 2022a), GEM (Du et al., 2021) and our DPF. GASP (Dupont et al., 2022b) employs a GAN paradigm: the generator produces a field while the discriminator operates on discrete points from the field, and distinguishes input source, *i.e.*, either real or generated.

In contrast to prior work (Dupont et al., 2022a; Du et al., 2021; Dupont et al., 2022b), we formulate a *diffusion generative model* over functions in a *single-stage* approach. This permits efficient end-to-end training without relying on an initial reconstruction stage or without tweaking the adversarial game, which we empirically find to lead to compelling results. Our contributions are summarized as follows:

- We introduce the Diffusion Probabilistic Field (DPF) which extends the formulation of diffusion generative models to field representations.

- We formulate a probabilistic generative model over fields in a single-stage model using an explicit field parametrization, which differs from recent work (Dupont et al., 2022a; Du et al., 2021) and simplifies the training process by enabling end-to-end learning.

- We empirically demonstrate that DPF can successfully capture distributions over functions across different domains like images, 3D geometry and spherical data, outperforming recent work (Dupont et al., 2022a; Du et al., 2021; Dupont et al., 2022b).

## 2 Background: Denoising Diffusion Probabilistic Models

Denoising Diffusion Probabilistic Models (DDPMs) belong to the broad family of latent variable models. We refer the reader to Everett (2013) for an in depth review. In short, to learn a parametric data distribution $p_\theta(x_0)$ from an empirical distribution of finite samples $q(x_0)$, DDPMs reverse a diffusion Markov Chain (*i.e.*, the forward diffusion process) that generates latents $x_{1:T}$ by gradually adding Gaussian noise to the data $x_0 \sim q(x_0)$ for $T$ time-steps as follows:

$$q(x_t|x_{t-1}) := \mathcal{N}\left(x_{t-1}; \sqrt{\bar{\alpha}_t}x_0, (1-\bar{\alpha}_t)\mathbf{I}\right). \tag{1}$$

Here, $\bar{\alpha}_t$ is the cumulative product of fixed variances with a handcrafted scheduling up to time-step $t$. Ho et al. (2020) highlight two important observations that make training of DDPMs efficient: i) Eq. (1) adopts sampling in closed form for the forward diffusion process. ii) reversing the diffusion

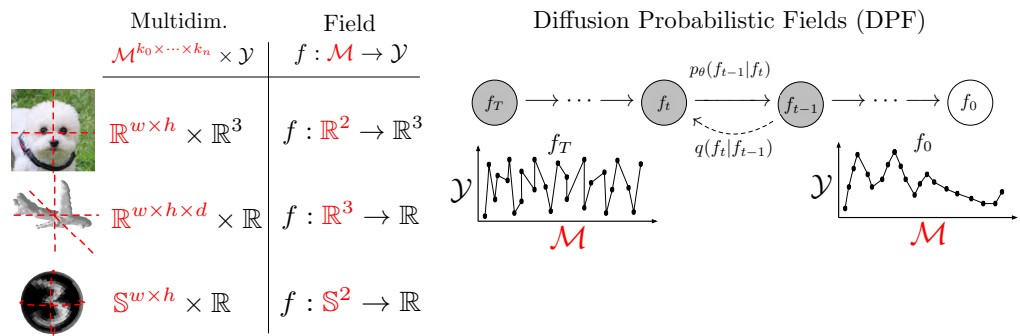

Figure 1: The left panel shows the parametrization of each data domain as a field. For visualization purposes we use the color red to denote the input to the field function (*e.g.*, a metric space $\mathcal{M}$ where the field is defined). In the right panel we show the graphical model of DPF, a diffusion generative model to capture distributions over fields. In DPF the latent variables are fields $f_{1:T}$ that can be evaluated continuously. By taking a field parametrization of data we unify diffusion generative modeling across different domains (images, 3D shapes, spherical images) using the same score network implementation.

process is equivalent to learning a sequence of denoising (or score) networks $\epsilon_\theta$, with tied weights. Reparametrizing Eq. (1) as $\boldsymbol{x}_t = \sqrt{\bar{\alpha}_t}\boldsymbol{x}_0 + \sqrt{1 - \bar{\alpha}_t}\epsilon$ results in the "simple" DDPM loss

$$\mathcal{L}_\theta = \mathbb{E}_{t\sim[0,T],\boldsymbol{x}_0\sim q(\boldsymbol{x}_0),\epsilon\sim\mathcal{N}(0,\mathbf{I})} \left[\|\epsilon - \epsilon_\theta(\sqrt{\bar{\alpha}_t}\boldsymbol{x}_0 + \sqrt{1 - \bar{\alpha}_t}\epsilon, t)\|^2\right], \qquad (2)$$

which makes learning of the data distribution $p_\theta(\boldsymbol{x}_0)$ both efficient and scalable.

At inference time, we compute $\boldsymbol{x}_0 \sim p_\theta(\boldsymbol{x}_0)$ via ancestral sampling (Ho et al., 2020). Concretely, we start by sampling $\boldsymbol{x}_T \sim \mathcal{N}(\mathbf{0},\mathbf{I})$ and iteratively apply the score network $\epsilon_\theta$ to denoise $\boldsymbol{x}_T$, thus reversing the diffusion Markov Chain to obtain $\boldsymbol{x}_0$. Sampling $\boldsymbol{x}_{t-1} \sim p_\theta(\boldsymbol{x}_{t-1}|\boldsymbol{x}_t)$ is equivalent to computing the update

$$\boldsymbol{x}_{t-1} = \frac{1}{\sqrt{\alpha_t}}\left(\boldsymbol{x}_t - \frac{1 - \alpha_t}{\sqrt{1 - \alpha_t}}\epsilon_\theta(\boldsymbol{x}_t, t)\right) + \mathbf{z}, \qquad (3)$$

where at each inference step a stochastic component $\boldsymbol{z} \sim \mathcal{N}(\mathbf{0},\mathbf{I})$ is injected, resembling sampling via Langevin dynamics (Welling & Teh, 2011).

A central part of the learning objective in Eq. (2) is the score network $\epsilon_\theta$, which controls the marginal distribution $p_\theta(\boldsymbol{x}_{t-1}|\boldsymbol{x}_t)$. Notably, score networks are heavily tailored for each specific data domain. For example, in the image domain, score networks are based on a UNet (Ronneberger et al., 2015) with multiple self-attention blocks (Nichol & Dhariwal, 2021). In contrast, for 3D structures like molecules, score networks are based on graph neural nets (Hoogeboom et al., 2022). To unify the design of score networks across data domains, in this paper, we present the diffusion probabilistic field (DPF), which introduces a unified formulation of the score network that can be applied to multiple domains by representing data samples as fields.

## 3 DIFFUSION PROBABILISTIC FIELDS

A diffusion probabilistic field (DPF) is a diffusion generative model that captures distributions over fields. We are given observations in the form of an empirical distribution $q(f_0)$ over fields (living in an unknown field manifold) where a field $f_0 : \mathcal{M} \to \mathcal{Y}$ maps elements from a metric space $\mathcal{M}$ to a signal space $\mathcal{Y}$. For example, in the image domain an image can be defined as a field that maps 2D pixel coordinates to RGB values $f_0 : \mathbb{R}^2 \to \mathbb{R}^3$. In DPF the latent variables $f_{1:T}$ are fields that can be continuously evaluated. To tackle the problem of learning a diffusion generative model over fields

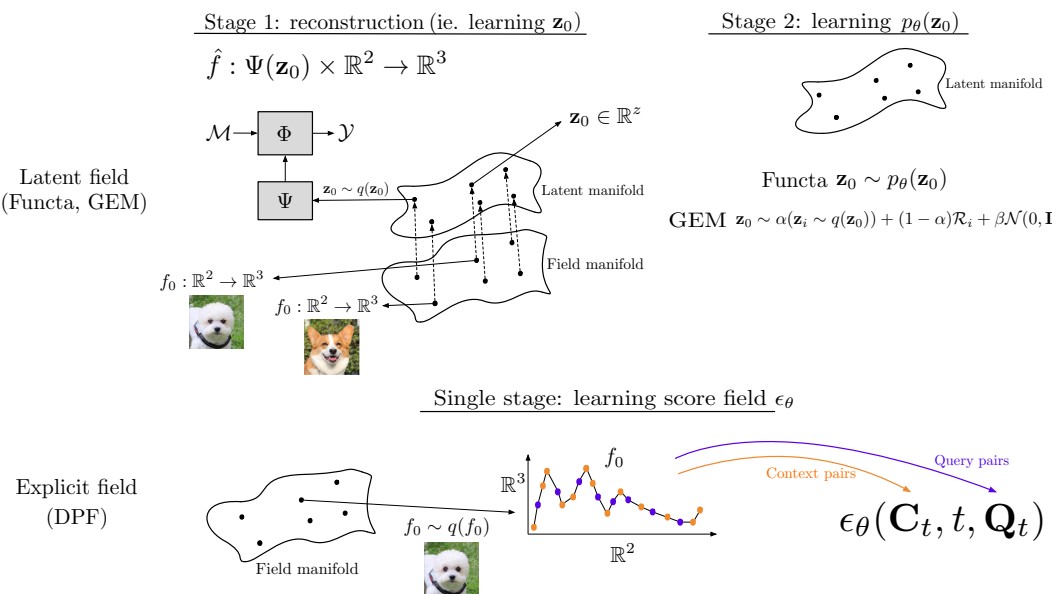

Figure 2: In DPF we explicitly parameterize a field by a set of coordinate-signal pairs, or *context pairs*, as opposed to a latent vector $z_0$ as in Functa (Dupont et al., 2022a) or GEM (Du et al., 2021). This explicit field parametrization allows us to side-step the reconstruction training stage of prior approaches and instead *directly model the distribution of fields* rather than the distribution of latents that encode fields.

we need to successfully deal with the infinite dimensional nature of the field representation in the forward process as well as in the score network and backward process.

We adopt an explicit field parametrization, where a field is characterized by a set of coordinate-signal pairs $\{(\boldsymbol{m}_c, \boldsymbol{y}_{(c,0)})\}$, $\boldsymbol{m}_c \in \mathcal{M}, \boldsymbol{y}_{(c,0)} \in \mathcal{Y}$, which we denote as *context pairs*. For clarity we row-wise stack context pairs and refer to the resulting matrix via $\mathbf{C}_0 = [\mathbf{M}_c, \mathbf{Y}_{(c,0)}]$. Here, $\mathbf{M}_c$ denotes the coordinate portion of all context pairs and $\mathbf{Y}_{(c,0)}$ denotes the signal portion of all context pairs at time $t = 0$. Note that the coordinate portion does not depend on the time-step by design.[2] This is a key difference with respect to Functa (Dupont et al., 2022a) or GEM (Du et al., 2021), both of which adopt a latent parametrization of fields, where a learnt field $\hat{f} : \Psi(\boldsymbol{z}_0) \times \mathcal{M} \to \mathcal{Y}$ is parametrized by a latent weight vector $\boldsymbol{z}_0$ through a hypernetwork model $\Psi$ (Ha et al., 2017). Using a latent parametrization forces a reconstruction stage in which latents $\boldsymbol{z}_0$ are first optimized to reconstruct their corresponding field (Park et al., 2019; Du et al., 2021) (*i.e.*, defining an empirical distribution of latents $q(\boldsymbol{z}_0)$ living in an unknown latent manifold). A prior $p_\theta(\boldsymbol{z}_0)$ over latents is then learnt *independently* in a second training stage (Dupont et al., 2022a). In contrast, our explicit field parametrization allows us to formulate a *score field network*, enabling DPF to directly model a distribution over fields, which results in improved performance (cf. Sect. 4). Fig. 2 depicts the differences between latent field parametrizations in Functa (Dupont et al., 2022a) and GEM (Du et al., 2021), and the explicit field parametrization in DPF.

Adopting an explicit field parametrization, we define the forward process for context pairs by diffusing the signal and keeping the coordinates fixed. Consequently the forward process for context pairs reads as follows:

$$\mathbf{C}_t = [\mathbf{M}_c, \mathbf{Y}_{(c,t)} = \sqrt{\bar{\alpha}_t}\mathbf{Y}_{(c,0)} + \sqrt{1 - \bar{\alpha}_t}\epsilon_c], \tag{4}$$

where $\epsilon_c \sim \mathcal{N}(\mathbf{0}, \mathbf{I})$ is a noise vector of the appropriate size. We now turn to the task of formulating a score network for fields. By definition, the score network needs to take as input the context pairs (*i.e.*, the field parametrization), and needs to accept being evaluated continuously in $\mathcal{M}$ in order to be

---

[2]We assume the geometry of fields does not change.

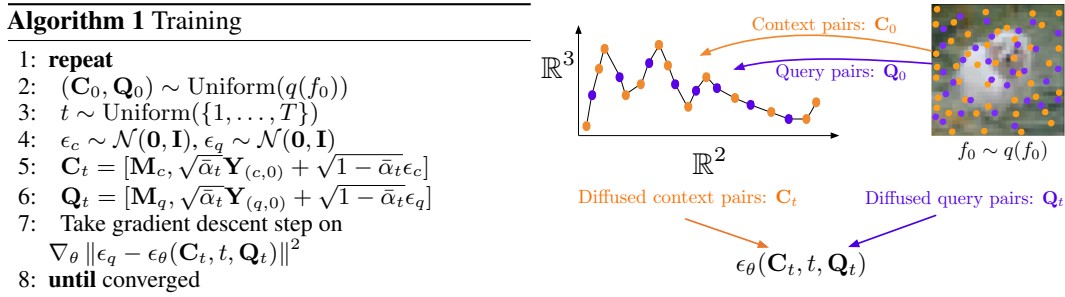

**Algorithm 1** Training

1: **repeat**
2: $\quad (\mathbf{C}_0, \mathbf{Q}_0) \sim \text{Uniform}(q(f_0))$
3: $\quad t \sim \text{Uniform}(\{1, \dots, T\})$
4: $\quad \epsilon_c \sim \mathcal{N}(\mathbf{0}, \mathbf{I}), \epsilon_q \sim \mathcal{N}(\mathbf{0}, \mathbf{I})$
5: $\quad \mathbf{C}_t = [\mathbf{M}_c, \sqrt{\bar{\alpha}_t}\mathbf{Y}_{(c,0)} + \sqrt{1 - \bar{\alpha}_t}\epsilon_c]$
6: $\quad \mathbf{Q}_t = [\mathbf{M}_q, \sqrt{\bar{\alpha}_t}\mathbf{Y}_{(q,0)} + \sqrt{1 - \bar{\alpha}_t}\epsilon_q]$
7: $\quad$ Take gradient descent step on
$\quad\quad \nabla_\theta \|\epsilon_q - \epsilon_\theta(\mathbf{C}_t, t, \mathbf{Q}_t)\|^2$
8: **until** converged

Figure 3: **Left:** DPF training algorithm. **Right**: Visual depiction of a training iteration for a field in the image domain. See Sect. 3 for definitions.

a field. We do this by using *query pairs* $\{\boldsymbol{m}_q, \boldsymbol{y}_{(q,0)}\}$. Equivalently to context pairs, we row-wise stack query pairs and denote the resulting matrix as $\mathbf{Q}_0 = [\mathbf{M}_q, \mathbf{Y}_{(q,0)}]$. Note that the forward diffusion process is equivalently defined for both context and query pairs:

$$\mathbf{Q}_t = [\mathbf{M}_q, \mathbf{Y}_{(q,t)} = \sqrt{\bar{\alpha}_t}\mathbf{Y}_{(q,0)} + \sqrt{1 - \bar{\alpha}_t}\epsilon_q], \tag{5}$$

where $\epsilon_q \sim \mathcal{N}(\mathbf{0}, \mathbf{I})$ is a noise vector of the appropriate size. However, the underlying field is solely defined by context pairs, and query pairs merely act as points on which to evaluate the score network. The resulting *score field* model is formulated as follows, $\hat{\epsilon}_q = \epsilon_\theta(\mathbf{C}_t, t, \mathbf{Q}_t)$.

The design space for the score field model is spans all architectures that can process data in the form of sets, like transformers or MLPs. In particular, efficient transformer architectures offer a straightforward way to deal with large numbers of context and query pairs, as well as good mechanism for query pairs to interact with context pairs via attention. For most of our experiments we use a PerceiverIO (Jaegle et al., 2022), an efficient transformer encoder-decoder architecture, see Fig. 11 and Sect. B for details. In addition, in Sect. D we show that other architectures like vanilla Transformer Encoders (Vaswani et al., 2017) and MLP-mixers (Tolstikhin et al., 2021) are also viable candidates offering a very flexible design space without sacrificing the generality of the formulation of DPF.

Using the explicit field characterization and the score field network, we obtain the training and inference procedures in Alg. 1 and Alg. 2, respectively, which are accompanied by illustrative examples for a field representation of images. For training, we uniformly sample context and query pairs from $f_0 \sim \text{Uniform}(q(f_0))$ and only corrupt their signal using the forward process in Eq. (4) and Eq. (5). We then train the score field network $\epsilon_\theta$ to denoise the signal in query pairs, given context pairs (see Fig. 11 for a visualization using a PerceiverIO implementation). During sampling, to generate a field $f_0 \sim p_\theta(f_0)$ we first define query pairs $\mathbf{Q}_T = [\mathbf{M}_q, \mathbf{Y}_{(q,T)} \sim \mathcal{N}(\mathbf{0}, \mathbf{I})]$ on which the field will be evaluated. Note that the number of points on which the score field is evaluated during sampling has to be fixed (*e.g.*, to generate an image with $32 \times 32 = 1024$ pixels we define 1024 query pairs $\mathbf{Q}_T$ at time $t = T$). We then let context pairs be a random subset of the query pairs. We use the context pairs to denoise query pairs and follow ancestral sampling as in the vanilla DDPM (Ho et al., 2020).[3] Note that during inference the coordinates of the context and query pairs do not change, only their corresponding signal value. The result of sampling is given by $\mathbf{Q}_0$ which is the result of evaluating $f_0$ at coordinates $\mathbf{M}_q$.

## 4 EXPERIMENTAL RESULTS

We present results on multiple domains: 2D image data, 3D geometry data, and spherical data. Across all domains we use the same score network architecture. We only adjust the dimensionality of the metric space $\mathcal{M}$ and the signal space $\mathcal{Y}$ as well as the network capacity (*i.e.*, number of layers, hidden

---

[3]More efficient sampling approaches like DDIM (Song et al., 2021a) are trivially adapted to DPF.

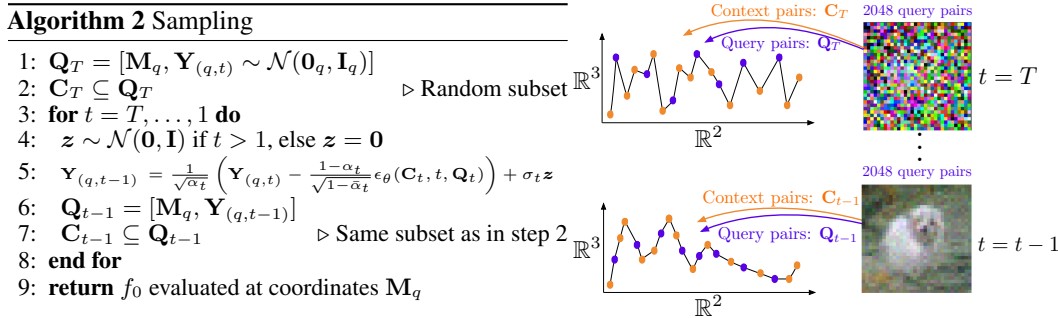

**Algorithm 2** Sampling

1: $\mathbf{Q}_T = [\mathbf{M}_q, \mathbf{Y}_{(q,t)} \sim \mathcal{N}(\mathbf{0}_q, \mathbf{I}_q)]$
2: $\mathbf{C}_T \subseteq \mathbf{Q}_T$ $\triangleright$ Random subset
3: **for** $t = T, \dots, 1$ **do**
4: $\quad \mathbf{z} \sim \mathcal{N}(\mathbf{0}, \mathbf{I})$ if $t > 1$, else $\mathbf{z} = \mathbf{0}$
5: $\quad \mathbf{Y}_{(q,t-1)} = \frac{1}{\sqrt{\alpha_t}} \left( \mathbf{Y}_{(q,t)} - \frac{1-\alpha_t}{\sqrt{1-\bar{\alpha}_t}} \epsilon_\theta(\mathbf{C}_t, t, \mathbf{Q}_t) \right) + \sigma_t \mathbf{z}$
6: $\quad \mathbf{Q}_{t-1} = [\mathbf{M}_q, \mathbf{Y}_{(q,t-1)}]$
7: $\quad \mathbf{C}_{t-1} \subseteq \mathbf{Q}_{t-1}$ $\triangleright$ Same subset as in step 2
8: **end for**
9: **return** $f_0$ evaluated at coordinates $\mathbf{M}_q$

Figure 4: **Left:** DPF sampling algorithm. **Right**: Visual depiction of the sampling process for a field in the image domain.

| CelebA-HQ $64^2$ | FID ↓ | Pre. ↑ | Rec. ↑ |
|---|---|---|---|
| VAE (Kingma & Welling, 2014) | 175.33 | 0.799 | 0.001 |
| StyleGAN2 (Karras et al., 2020a) | 5.90 | 0.618 | 0.481 |
| Functa (Dupont et al., 2022a) | 40.40 | 0.577 | 0.397 |
| GEM (Du et al., 2021) | 30.42 | 0.642 | 0.502 |
| GASP (Dupont et al., 2022b) | 13.50 | 0.836 | 0.312 |
| DPF (ours) | 13.21 | 0.866 | 0.347 |

| CIFAR-10 $32^2$ | FID ↓ | IS ↑ |
|---|---|---|
| PixelIQN (Ostrovski et al., 2018) | 49.46 | 5.29 |
| NCSNv2 (Song & Ermon, 2020) | 31.75 | - |
| StyleGAN2 (Karras et al., 2020a) | 3.26 | 9.74 |
| DDPM (Ho et al., 2020) | 3.17 | 9.46 |
| GEM (Du et al., 2021) | 23.83 | 8.36 |
| DPF (ours) | 15.10 | 8.43 |

Table 1: **Quantitative evaluation of image generation** on CelebA-HQ (Karras et al., 2018). The middle bar separates domain-specific (*top*) from domain-agnostic approaches (*bottom*).

Table 2: **Quantitative evaluation of image generation** on CIFAR-10 (Krizhevsky, 2009).The middle bar separates domain-specific (*top*) from domain-agnostic approaches (*bottom*).

units per layer, etc.) when needed. We implement the field score network $\epsilon_\theta$ using a PerceiverIO architecture (Jaegle et al., 2022), an efficient transformer that enables us to scale the number of both context pairs and query pairs. Additional architecture hyperparameters and implementation details are provided in the appendix.

## 4.1 2D IMAGES

We present empirical results on two standard image benchmarks: CelebA-HQ (Karras et al., 2018) $64^2$ and CIFAR-10 (Krizhevsky, 2009) $32^2$. All image datasets are mapped to their field representation, where an image $\boldsymbol{x} \in \mathbb{R}^{h \times w \times 3}$ is represented as a function $f : \mathbb{R}^2 \to \mathbb{R}^3$ defined by coordinate-RGB pairs. In Tab. 1 and Tab. 2 we compare DPF with Functa (Dupont et al., 2022a), GEM (Du et al., 2021), and GASP (Dupont et al., 2022b), which are domain-agnostic generative models that employ field representations. For completeness, we also report results for the domain-specific VAE (Kingma & Welling, 2014), StyleGAN2 (Karras et al., 2020a) and DDPM (Ho et al., 2020). Similarly, on CIFAR-10 data we compare with multiple domain-specific methods including auto-regressive (Ostrovski et al., 2018) and score-based (Song & Ermon, 2020) approaches. We report Fréchet Inception Distance (FID) (Heusel et al., 2017), Inception Score (IS) (Salimans et al., 2016) and precision/recall metrics (Sajjadi et al., 2018) for different datasets.

We observe that DPF obtains compelling generative performance on both CelebA-HQ $64^2$ and CIFAR-10 data. In particular, DPF outperforms recent approaches that aim to unify generative modeling across different data domains such as Functa (Dupont et al., 2022a) and GEM (Du et al., 2021). In particular, we observe that DPF obtains the best Precision score across all domain-agnostic approaches on CelebA-HQ $64^2$ (Karras et al., 2018), as well as very competitive FID scores. One interesting finding is that GASP Dupont et al. (2022b) reports comparable FID score than DPF. However, when inspecting qualitative examples shown in Fig. 5 we observe GASP samples to contain artifacts typically obtained by adversarial approaches. In contrast, samples generated from DPF are more coherent and without artifacts. We believe the texture diversity caused by these artifacts to be the reason for the difference in FID scores between GASP and DPF (similar findings were discussed

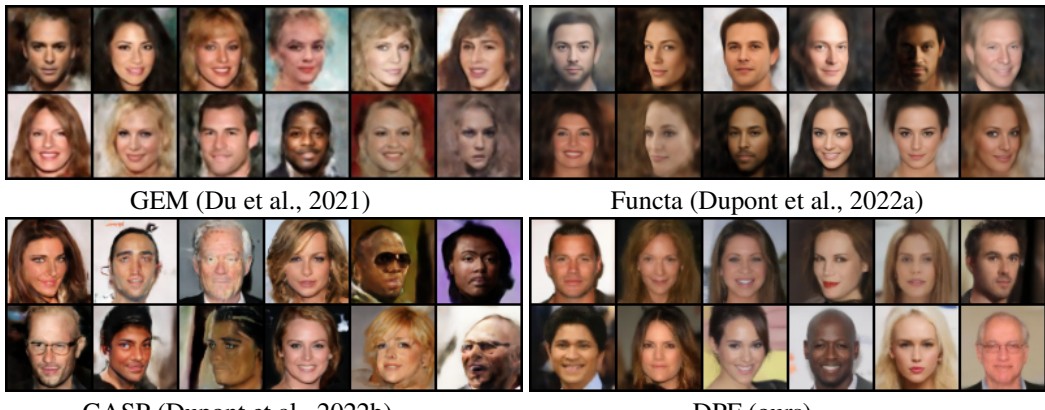

GEM (Du et al., 2021)  ·  Functa (Dupont et al., 2022a)

GASP (Dupont et al., 2022b)  ·  DPF (ours)

Figure 5: **Qualitative comparison of different domain-agnostic approaches** on CelebA-HQ $64^2$ (Karras et al., 2018). DPF generates diverse and globally consistent samples, without the background blurriness of two-stage approaches like Functa (Dupont et al., 2022a) and GEM (Du et al., 2021) that rely on an initial reconstruction step, or the artifacts of adversarial approaches like GASP (Dupont et al., 2022b). All the images in this figure have been generated by a model, *i.e.*, we do not directly reproduce images from the CelebA-HQ dataset (Karras et al., 2018).

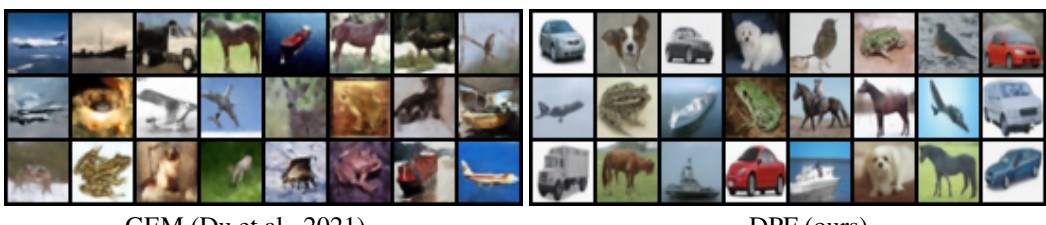

GEM (Du et al., 2021)  ·  DPF (ours)

Figure 6: **Qualitative comparison** of GEM (Du et al., 2021) and DPF on CIFAR-10 (Krizhevsky, 2009). DPF generates crisper samples than GEM (Du et al., 2021) while also better capturing the empirical distribution.

in (Dupont et al., 2022a)). Finally, when studying empirical results on CIFAR-10 shown in Tab. 2, we again see that DPF performs better than GEM (Du et al., 2021) both in terms of FID and IS.

Notably, we can observe a gap between domain-agnostic and the most recent domain-specific approaches such as StyleGAN2 (Karras et al., 2020a) and DDPM (Ho et al., 2020). This gap is a result of two main factors. First, domain-specific approaches can incorporate design choices tailored to their domain in the score network (*i.e.*, translation equivariance in CNNs for images). Second, the training and inference settings are typically different for the domain-agnostic and domain-specific approaches reported in Tab. 1 and Tab. 2. To further study this issue we refer readers to Sect. A, where we compare the performance of DPF and DDPM (Ho et al., 2020) using the same training and evaluation settings.

Finally, we show qualitative generation results for domain-agnostic approaches on CelebA-HQ $64^2$ (Karras et al., 2018) and CIFAR-10 (Krizhevsky, 2009) in Fig. 5 and Fig. 6, respectively. We note that for CelebA-HQ, DPF generates diverse and globally consistent samples, without the blurriness (particularly in the backgrounds) observed in two-stage approaches (Dupont et al., 2022a; Du et al., 2021) that rely on an initial reconstruction step, or the artifacts of adversarial approaches (Dupont et al., 2022b). When evaluated on CIFAR-10 (Krizhevsky, 2009), we see that DPF generates crisper and more diverse results than GEM (Du et al., 2021).

### 4.2 3D GEOMETRY

We now turn to the task of modeling distributions over 3D objects and present results on the ShapeNet dataset (Chang et al., 2015). We follow the settings in GEM (Du et al., 2021) and train our model on

| ShapeNet $64^3$ | Coverage ↑ | MMD ↓ |
|---|---|---|
| Latent GAN (Chen & Zhang, 2019) | 0.389 | 0.0017 |
| GASP (Dupont et al., 2022b) | 0.341 | 0.0021 |
| GEM (Du et al., 2021) | 0.409 | 0.0014 |
| DPF (ours) | 0.419 | 0.0016 |

Table 3: **Quantitative evaluation of 3D geometry generation** on ShapeNet (Chang et al., 2015). DPF outperforms prior approaches both in terms of Coverage and MMD metrics.

$\sim 35$k ShapeNet objects, where each object is represented as a voxel grid at a $64^3$ resolution. Each object is then represented as a function $f : \mathbb{R}^3 \to \mathbb{R}^1$. Following the settings of GEM (Du et al., 2021), we report coverage and MMD metrics (Achlioptas et al., 2018) computed from sampling 2048 points on the meshes obtained from the ground truth and generated voxel grids. We compare them using the Chamfer distance. In Tab. 3 we compare DPF performance with 3 baselines: Latent GAN (Chen & Zhang, 2019), GASP (Dupont et al., 2022b) and GEM (Du et al., 2021). We train DPF at $32^3$ resolution. During sampling we evaluate $32^3$ query pairs on the score field, then reshape the results to a 3D grid of $32^3$ resolution and tri-linearly up-sample to the final $64^3$ resolution for computing evaluation metrics.

DPF outperforms both GEM (Du et al., 2021) and GASP (Dupont et al., 2022b) in learning the multimodal distribution of objects in ShapeNet, as shown by the coverage metric. In addition, while GEM (Du et al., 2021) performs better in terms of MMD, we do not observe this difference when visually comparing the generated samples shown in Fig. 7. We attribute this difference in MMD scores to the fact that MMD over-penalizes fine-grained geometry.

## 4.3 Data on $\mathbb{S}^2$

Straightforwardly, DPF can also learn fields that are not defined in the Euclidean plane. To demonstrate this, we show results on signals defined over the sphere. In particular, following Cohen et al. (2018), we use a stereographic projection to map image data onto the sphere. Hence, each resulting example is represented as a field $f_0 : \mathbb{S}^2 \to \mathbb{R}^d$. To uniformly sample points in $\mathbb{S}^2$ we use the Driscoll-Healy algorithm (Driscoll & Healy, 1994) and sample points at a resolution of $32^2$ and $64^2$ for spherical MNIST (LeCun et al., 1998) and AFHQ (Choi et al., 2020) data, respectively. In Fig. 8 we show the distribution of real examples for MNIST (LeCun et al., 1998) and AFHQ (Choi et al., 2020) images projected on the sphere as well as the samples generated by our DPF. Unsurprisingly, since DPF is agnostic to the geometry of the metric space $\mathcal{M}$, it can generate crisp and diverse samples for fields defined on the sphere.

## 5 Related Work

Generative modeling has advanced significantly in recent years with generative adversarial nets (GANs) (Goodfellow et al., 2014; Mao et al., 2017; Karras et al., 2020b) and variational auto-encoders (VAEs) (Vahdat & Kautz, 2020) showing impressive performance. Even more recently, diffusion-based generative modeling has obtained remarkable compelling results (Dickstein et al., 2015; Ho et al., 2020; Song et al., 2021b).

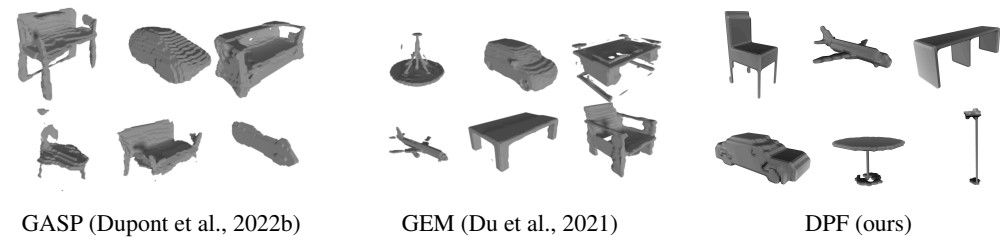

GASP (Dupont et al., 2022b)     GEM (Du et al., 2021)     DPF (ours)

Figure 7: **Qualitative comparison of different domain-agnostic approaches** on ShapeNet (Chang et al., 2015).

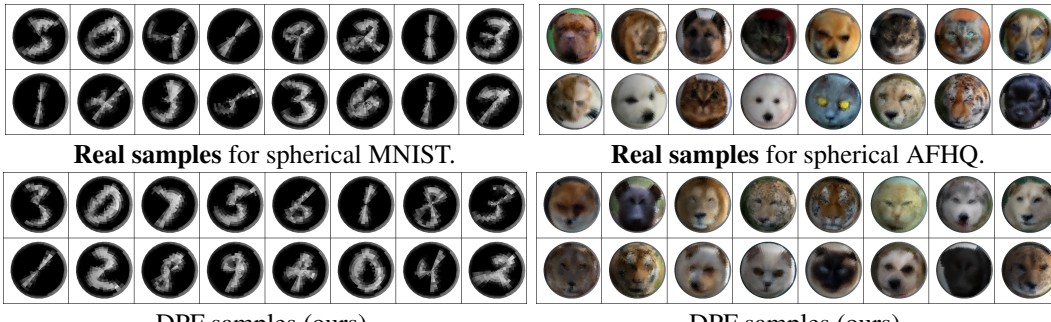

**Figure 8: Qualitative comparison of empirical and generated samples** for spherical versions of MNIST and AFHQ (Choi et al., 2020).

The formulation developed in DPF is orthogonal to the body of work on Riemannian generative models (Bortoli et al., 2022; Gemici et al., 2016; Rozen et al., 2021). The goal in Riemannian generative modeling is to explicitly constraint the learned density to a Riemannian manifold structure. For example, a Riemannian generative model can learn a distribution of points $x \in \mathbb{S}^2$ on the 2D sphere $\mathbb{S}^2$, explicitly enforcing that any generated samples lie on the sphere. In contrast, DPF learns a generative model over a distribution of multiple samples of signals defined on the sphere, or any other metric space.

The DPF formulation also differs from the recently introduced Functa (Dupont et al., 2022a), GEM (Du et al., 2021) and GASP (Dupont et al., 2022b). The first two approaches adopt a latent field parametrization (Park et al., 2019) and a two-stage training paradigm. However, different from our work, the field network in Functa and GEM is parametrized via a hypernetwork (Ha et al., 2017) that takes as input a trainable latent vector. During training, a small latent vector for each field is optimized in an initial auto-decoding or compression) stage (Park et al., 2019). In the second stage, a probabilistic model is learned on the latent vectors. GASP (Dupont et al., 2022b) leverages a GAN whose generator produces field data whereas a point cloud discriminator operates on discretized data and aims to differentiate input source, *i.e.*, either real or generated. Two-stage approaches like Functa (Dupont et al., 2022a) or GEM (Du et al., 2021) make training the probabilistic model in the second stage more computationally efficient than DPF. This training efficiency of the probabilistic model often comes at the cost of compressing fields into small latent vectors in the first stage, which has a non-negligible computational cost, specially for large datasets of fields.

The formulation introduced in DPF has is closely related to recent work on Neural Processes (NPs) (Garnelo et al., 2018; Kim et al., 2019; Dutordoir et al., 2022), which also learn distributions over functions via context and query pairs. As opposed to the formulation of Neural Processes which optimizes an ELBO Kingma & Welling (2014) we formulate DPF as a denoising diffusion process in function space, which results in a robust denoising training objective and a powerful iterative inference process. In comparison with concurrent work in Neural Processes (Garnelo et al., 2018; Kim et al., 2019; Dutordoir et al., 2022) we do not explicitly formulate a conditional inference problem and look at the more challenging task of learning an unconditional generative model over fields. We extensively test our hypothesis on complex field distributions for 2D images and 3D shapes, and on distribution of fields defined over non-Euclidean geometry.

## 6 CONCLUSION

In this paper we made progress towards modeling distributions over fields by introducing DPF. A diffusion probabilistic model that directly captures a distribution over fields without resorting to a initial reconstruction stage (Dupont et al., 2022a; Du et al., 2021) or tweaking unstable adversarial approaches (Dupont et al., 2022b). We show that DPF can capture distributions over fields in different domains without making any assumptions about the data. This enables us to use the same denoising architecture across domains obtaining satisfactory results. In addition, we show that DPF can be used to learn distributions over fields defined on non-Euclidean geometry.

## 7 ETHICAL STATEMENT

When considering ethical impact of generative models like DPF a few aspects that need attention are the use generative models for creating disingenuous data, *e.g.*, "DeepFakes" Mirsky & Lee (2021), training data leakage and privacy Tinsley et al. (2021), and amplification of the biases present in training data Jain et al. (2020). For an in-depth review of ethical considerations in generative modeling we refer the reader to Rostamzadeh et al. (2021). In addition, we want to highlight that any face images we show are generated from our model. We do not directly reproduce face images from any dataset.

## 8 REPRODUCIBILITY STATEMENT

We take great care in the reproducibility of our results. We provide links to the public implementations that can be used to replicate our results in Sect. A and Sect. B, as well as describing all training parameters in Tab. 6. All of the datasets we report results on are public and can be freely downloaded.

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

## APPENDIX

In this supplementary material, we provide an ablation study (Sect. A), implementation details (Sect. B), additional experiments and visualizations (Sect. C), limitations and future work (Sect. G).

## A    ABLATION

In this section we ablate the critical design choices and hyperparameters of DPF. We train all model variants on the CelebA-HQ $64^2$ dataset (Karras et al., 2018).

**Performance gap between DPF and DDPM (Ho et al., 2020).** We aim to provide a fair comparison between DPF and DDPM (Ho et al., 2020). For this we use the implementation of the DDPM score network by Ho et al. (2020)[4] and train both DDPM and DPF with the same settings: batch size, learning rate schedule, number of epochs, etc. During inference we use ancestral sampling and perform 1000 denoising steps. In Tab. 4 we observe the gap between DDPM (Ho et al., 2020) (a domain-specific approach) and DPF to be much smaller than it is in Tab. 2. Furthermore, when we approximately equate the number of parameters of the score network in DDPM and our score field model (67.6M vs. 62.4M parameters), we find that DPF outperforms DDPM in both FID and recall metrics. We attribute this to the fact that DPF better captures globally consistent samples and long-range interactions. We show qualitative results generated by DDPM (Ho et al., 2020) and DPF in Fig. 9.

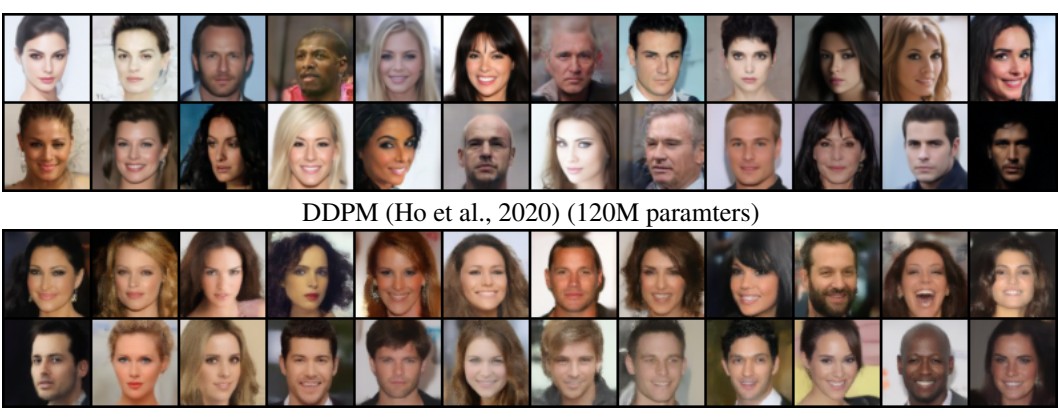

DDPM (Ho et al., 2020) (120M paramters)

DPF (ours, 62.4M paramters)

Figure 9: **Visualization of DDPM (Ho et al., 2020) and DPF results** on CelebA-HQ (Karras et al., 2018) dataset. All the images show in this figure are generated by a model, we do not directly reproduce images from the CelebA-HQ dataset (Karras et al., 2018)

Moreover, we are interested in understanding whether the major bottleneck for DPF is on encoding context pairs or decoding query pairs (*e.g.*, evaluating the field at query coordinates). To verify this, we design two ablation experiments. One experiment is to replace the DPF encoder with a UNet and the other one is to replace the DPF decoder with a UNet. We show quantitative results in Tab. 4. We observe that DPF with a UNet decoder achieves the best performance w.r.t. FID, precision and recall scores. These empirical results suggest that further improvements might be obtained by better decoding of query pairs in DPF.

**Resolution-free generation.** Importantly, DPF differs from classic DDPM (Ho et al., 2020) in that DPF generates fields that can be evaluated at continuous points using query pairs, *i.e.*, context pairs used during training differ from query pairs applied at inference. This enables DPF to learn continuous fields rather than naively memorizing discretized data samples. To verify this, we sample DPF at a different resolution than the one it was trained on. Specifically, we train DPF on CelebA-HQ (Karras et al., 2018) at a resolution of $64^2$ while using $128^2$ query pairs to generate high resolution images, as shown in Fig. 10. From Fig. 10, we observe that DPF can generate accurate high-resolution image fields given low-resolution context and query pairs during training. For context, we compare

---

[4]Code available at https://github.com/rosinality/denoising-diffusion-pytorch.

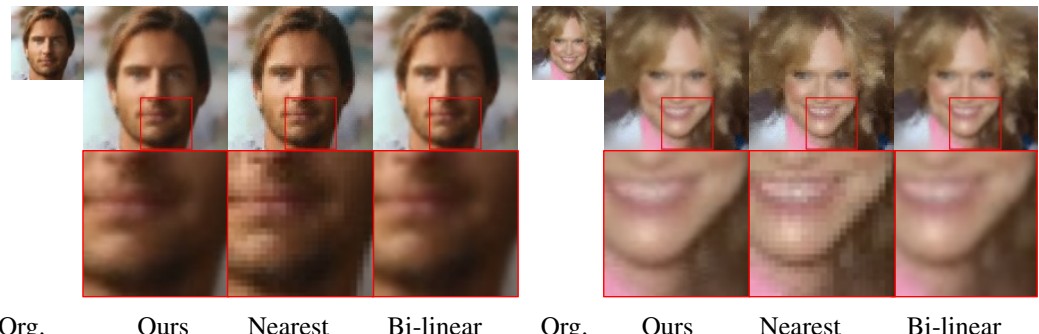

| Org. | Ours | Nearest | Bi-linear | Org. | Ours | Nearest | Bi-linear |

Figure 10: **Qualitative results of super-resolution image generation.** We generate images at $128^2$ resolution *(row 1)* given $64^2$ context points *(top left)*. We show zoom-in results *(row 2)*. We compare our results with nearest neighbor and bi-linear upsampling. All the images in this figure have been generated by a model, *i.e.*, we do not directly reproduce images from the CelebA-HQ dataset (Karras et al., 2018).

| CelebA-HQ $64^2$ | # Params. | # Channel | FID ↓ | Precision ↑ | Recall ↑ |
|---|---|---|---|---|---|
| DDPM (Ho et al., 2020) | 120M | 128 | 11.26 | 0.916 | 0.384 |
| DDPM (Ho et al., 2020) | 67.6M | 96 | 24.93 | 0.927 | 0.213 |
| DPF (ours) | 62.4M | - | 13.21 | 0.866 | 0.347 |
| DPF (w/ a UNet encoder) | 127M | 128 | 20.79 | 0.916 | 0.323 |
| DPF (w/ a UNet decoder) | 62.5M | 128 | 12.27 | 0.877 | 0.457 |

Table 4: **DDPM (Ho et al., 2020) vs. DPF under the same setting.** The DDPM models are implemented using a UNet architecture (Ronneberger et al., 2015) with a convolutional channel number of 128 *(row 1)* and 96 *(row 2)*, respectively. We also use a UNet architecture to replace either the encoder *(row 4)* or the decoder *(row 5)* of DPF.

our results with standard nearest neighbor or bi-linear upsampling. In Fig. 10, we show that our results enjoy comparable or higher quality w.r.t. texture details than the upsampled results.

**DPF hyperparameters.** We provide an ablation over different DPF hyperparameters. In Tab. 5 we study the effect of varying the number of context pairs used during training. In Tab. 6 we evaluate different PerceiverIO (Jaegle et al., 2022) hyperparameters. For the comparisons in Tab. 5 and Tab. 6 we train models for 180 epochs and compute FID metrics using $1k$ samples.

As we see in Tab. 5, decreasing the number of context pairs leads to a drop in FID scores. This is expected since a smaller number of context pairs reduces the fidelity at which the field is represented. Also noteworthy, we have not optimized score network hyper-parameters for a reduced number of context pairs, *i.e.*, they are selected only based on the performance of the model trained with the full set of context pairs.

The results reported in Tab. 6 show the effects of reducing the number of parameters in the PerceiverIO architecture, using different knobs. We observe: decreasing the capacity of the latents (either number of latents or their dimensionality) has a higher impact than decreasing the depth of the model (number of blocks in the encoder, decoder, or the number of self-attention layers in encoder blocks).

| CelebA-HQ $64^2$ | FID |
|---|---|
| # context pairs = 1028 (50%) | 103.86 |
| # context pairs = 2048 (75%) | 91.12 |
| # context pairs = 4096 (100%) | 74.89 |

Table 5: **Performance of DPF trained with different number of context pairs**. Context pairs are randomly samples during training. The FID score is evaluated after 180 epochs for $1k$ samples.

| **CelebA-HQ** $64^2$ | # Params. | FID |
|---|---|---|
| # latents = 256 | 62.3M | 93.54 |
| # dim latents = 256 | 22.8M | 96.63 |
| # decoder blocks = 2 | 59.7M | 77.29 |
| # blocks = 6 | 35.4M | 83.87 |
| # S.A. block = 1 | 43.5M | 75.52 |
| Base | 62.4M | 74.89 |

Table 6: **Performance of DPF with different PerceiverIO (Jaegle et al., 2022) hyperparameters.** The base hyperparameters we use in the experiments are: 512 latents with a dimensionality of 512, 12 encoder blocks with 2 self-attention modules and 4 decoder blocks. The FID score is evaluated after 180 epochs for $1k$ samples. For more details on hyperparameters and implementation details see Sect. B.

## B    IMPLEMENTATION DETAILS

The design space for the implementation of the score field in DPF spans all architectures that can deal with irregularly sampled data, like Transformers (Vaswani et al., 2017) or MLPs (Tolstikhin et al., 2021) (see Sect. D). Unless otherwise noted, we implement DPF using a PerceiverIO (Jaegle et al., 2022), an efficient encoder-decode transformer architecture. Our motivation to choose a PerceiverIO is that it provides a straightforward way of dealing with a large number context and query pairs, and a natural way for encoding the interaction between context and query pairs via attention.In Fig. 11 we show how context and query pairs are used within the PerceiverIO architecture. Specifically, the encoder maps context pairs to latent arrays (*e.g.* a set of learnable vectors) via a cross-attention layer. Similarly the decoder maps query pairs to latent arrays via cross-attention blocks. We refer readers to Jaegle et al. (2022) for additional discussions around the PerceiverIO architecture.

For conditioning the score computation on the time-step $t$ we simply concatenate a positional embedding representation of $t$ to the context and query pairs. PerceiverIO settings for all experiments with quantitative evaluation are show in Tab. 6. In practice, the DPF encoder has 12 transformer blocks, each of which contains 1 cross-attention layer followed by 2 self-attention layers. Both cross-attention and self-attention layers have 4 attention heads. The latent array has a dimension of $512 \times 512$ for $64^2$ resolution images and $1024 \times 1024$ for $32^2$ resolution images. We use Fourier feature position encoding to represent coordinates and time steps with a frequency of 10 and 64, respectively. We use an Adam (Kingma & Ba, 2015) optimizer during training. We set the learning rate to $1e - 4$. We set the batch size to 16 for all image datasets. We use 8 A100 GPUs for all experiments. Finally, for PerceiverIO we use a modification of the public repository.[5]

## C    ADDITIONAL EXPERIMENTS

**Visualization of denoising process.** In Fig. 12 we visualize the denoising process over time from left to right on CelebA-HQ (Karras et al., 2018) and CIFAR-10 (Krizhevsky, 2009) data.

**Visual results on additional image datasets.** In Fig. 16 we show qualitative results of DPF on additional image datasets, including AHFQ (Choi et al., 2020), FFHQ (Karras et al., 2019) and LSUN church (Yu et al., 2015) data at $32 \times 32$ resolution. In Fig. 17 we show videos of generated fields over spherical geometry (view with Adobe Acrobat to see animations).

## D    ARCHITECTURE ABLATION FOR THE SCORE FIELD NETWORK $\epsilon_\theta$ IN DPF

The formulation of DPF is independent of the specific implementation of the score network $\epsilon_\theta$, with the design space of the score model spanning a large number options. In particular, all architectures

---

[5]https://huggingface.co/docs/transformers/model_doc/perceiver

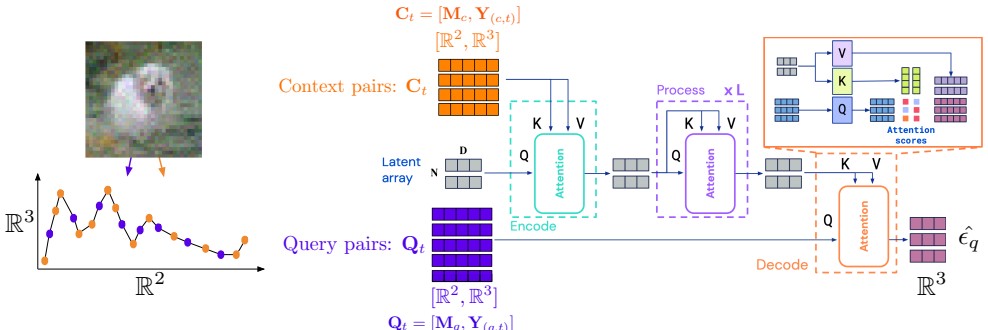

Figure 11: Interaction between context and query pairs in the PerceiverIO architecture. Context pairs $\mathbf{C}_t$ attend to a latent array of learnable parameters via cross attention. The latent array then goes through several self attention blocks. Finally, the query pairs $\mathbf{Q}_t$ cross-attend to the latent array to produce the final noise prediction $\hat{\epsilon}_q$.

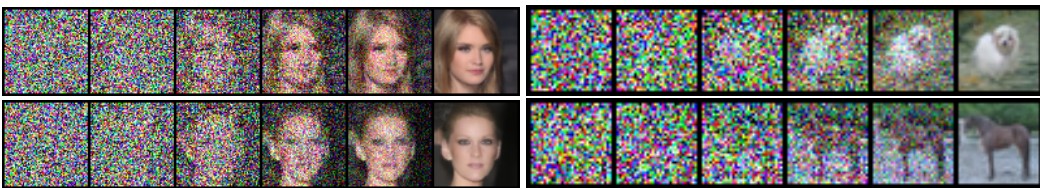

Figure 12: **Visualization of denoising process** from $t = 1000$ (*noise*) to $t = 0$ (*images*) on CelebA-HQ (Karras et al., 2018) (*left*) and CIFAR-10 (Krizhevsky, 2009) (*right*).

that can process irregular data like transformers or MLPs are viable candidates. To support this claim, we perform an evaluation on CelebA-HQ at $32 \times 32$ resolution where we compare 3 different architectures: a PerceiverIO Jaegle et al. (2022), a vanilla Transformer Encoder Vaswani et al. (2017) and an MLP-mixer (Tolstikhin et al., 2021). We report FID and KID metrics using 1000 samples obtained using the inference process described in Alg. 2.To provide a fair comparison we roughly equate the number of parameters (to approximately 55M) and settings (*e.g.* number of blocks, parameters per block, etc.) of each model and train them for 500 epochs. To simplify the evaluation we use the same number of context and query pairs. Both the Transformer Encoder and the MLP-mixer process context pairs as input using their respective architectures and the resulting latents are then concatenated with the corresponding query pairs and fed to linear projection layer for the final prediction.

In Tab. 8, we see how the formulation of DPF is amenable to different architecture implementations of the score field model. In particular, we observe relatively similar performance across the board for different architectures, from transformer based to MLPs. We also observe the same behaviour when looking at the qualitative samples shown in Fig. 14. This supports our claim that the benefits of DPF are due to its formulation and not the particular incarnation of the score field model. Each particularly architecture has its own benefits, for example, MLP-mixers allow for high throughput, transformer encoders are simple to implement, and PerceiverIO allows for large and variable number of context and query pairs. We believe that combining the strengths of these different architectures is a very promising future direction of progress for DPF. Finally, note that these empirical results are not directly comparable to ones reported in other sections of the paper since these models generally have roughly $50\%$ of the parameters of the models used in other sections.

Furthermore, in Fig. 13 we also provide a qualitative comparison of MLP-mixers and PerceiverIO results on the Spherical MNIST dataset used in Sect. 4. In which we can see how both a PerceiverIO and an MLP-mixer successfully capture the distribution of digits over the sphere. This supports the claim that the data-agnostic capabilities for generative modeling introduced in DPF are not a function of the particular implementation of the score field, but due to the formulation.

| Hyper-parameter | CelebA-HQ (Karras et al., 2018) | CIFAR-10 (Krizhevsky, 2009) | ShapeNet (Chang et al., 2015) |
|---|---|---|---|
| train res. | $64^2$ | $32^2$ | $32^3$ |
| eval res. | $64^2$ | $32^2$ | $64^3$ |
| #context pairs | 4096 | 2048 | 32768 |
| #query pairs | 4096 | 2048 | 32768 |
| #dim coordinates | 2 | 2 | 3 |
| #dim signal | 3 | 3 | 1 |
| #freq pos. embed | 10 | 10 | 10 |
| #freq pos. embed $t$ | 64 | 64 | 64 |
| #latents | 1024 | 512 | 512 |
| #dim latents | 1024 | 512 | 1024 |
| #blocks | 12 | 12 | 12 |
| #dec blocks | 1 | 1 | 1 |
| #self attends per block | 2 | 2 | 2 |
| #self attention heads | 4 | 4 | 4 |
| #cross attention heads | 4 | 4 | 4 |
| batch size | 128 | 128 | 16 |
| lr | $1e-4$ | $1e-4$ | $1e-4$ |
| epochs | 1300 | 1300 | 1000 |

Table 7: Hyperparameters and settings for DPF on different datasets.

| **CelebA-HQ** $32^2$ | FID ↓ | KID ↓ $\times 10^3$ |
|---|---|---|
| PeceiverIO (Jaegle et al., 2022) | 64.8 | 7.10 |
| Transf. Enc. (Vaswani et al., 2017) | 70.1 | 10.27 |
| MLP-mixer (Tolstikhin et al., 2021) | 66.8 | 7.89 |

Table 8: **Quantitative evaluation of image generation** on CelebA-HQ (Karras et al., 2018) at 32x32 resolution for different implementation of the score field $\epsilon_\theta$.

# E  QUALITATIVE COMPARISON WITH FUNCTA (DUPONT ET AL., 2022A) AND GASP (DUPONT ET AL., 2022B) ON $\mathbb{S}^2$.

We provide a qualitative comparison with Functa (Dupont et al., 2022a) and GASP (Dupont et al., 2022b) on fields defined over the sphere. In order to do this, we used the ERA5 temperature dataset released with GASP `https://github.com/EmilienDupont/neural-function-distributions` which is available at a $46 \times 90$ spherical resolution. It is important to note that this dataset is different from the ERA5 dataset used to train Functa (Dupont et al., 2022a), which is sampled at a higher resolution ($181 \times 360$) and is not publicly available in their repository `https://github.com/deepmind/functa`. In Fig. 15 we show samples generated by Functa (Dupont et al., 2022a), GASP (Dupont et al., 2022b) and DPF. We observe that samples generated by DPF are visually indistinguishable from those of Functa or GASP. DPF captures the same patterns as Functa and GASP, where the poles having cooler temperatures (*e.g.*. dark purple tones) and the equator and tropics having considerably high temperatures shown by the lighter yellow color.

# F  SAMPLING CONTEXT AND QUERY PAIRS

A natural question that falls from the formulation of DPF via context and query pairs is the following: *is there an intuition for which context and query pairs to sample from the field?*

For *query pairs* the answer is simple: one should sample as many pairs as required in the problem definition. For example, if the task is that of sampling a field representation of an image at $32 \times 32$ resolution, one should sample $32 \times 32 = 1024$ query pairs. In this sense, the number and distribution of query pairs is entirely problem dependent and flexible and DPF allows for this flexibility.

Selecting which *context pairs* to sample is a far more intricate problem. To discuss it we exploit the intimate relationship between the formulation in DPF and Gaussian Processes (GPs). In particular, one can look at the problem of choosing context pairs as equivalent to the problem of selecting *inducing variables* in Sparse GPs Quinonero-Candela & Rasmussen (2005), which is a combinatorial problem for which approximations are computed. While there exists different approximations to the

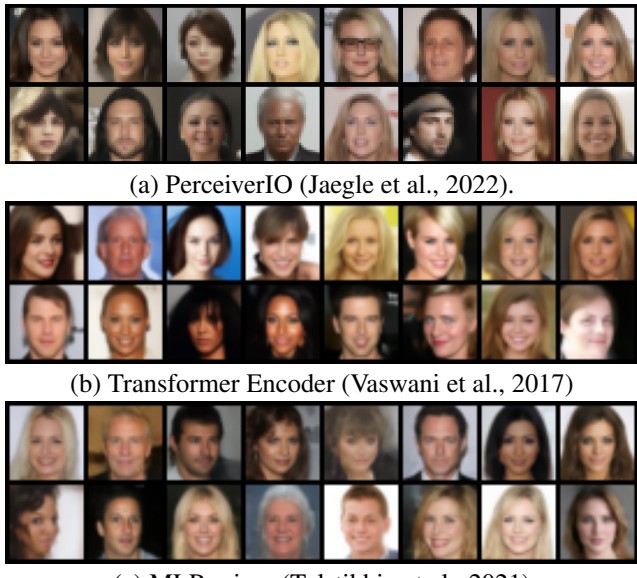

(a) PerceiverIO (Jaegle et al., 2022).

(b) Transformer Encoder (Vaswani et al., 2017)

(c) MLP-mixer (Tolstikhin et al., 2021)

Figure 13: **Qualitative comparison** of different architectures to implement the score field model $\epsilon_\theta$.

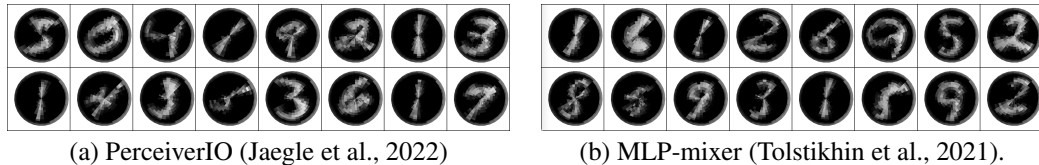

(a) PerceiverIO (Jaegle et al., 2022)          (b) MLP-mixer (Tolstikhin et al., 2021).

Figure 14: **Qualitative comparison** of different implementations of the score field for functions defined on the sphere. (a) PerceiverIO (Jaegle et al., 2022). (b) MLP-mixer (Tolstikhin et al., 2021).

problem of selecting inducing variables we highlight 3 classical methods which are representative broader classes of approaches. In (Titsias, 2009) a variational approximation that jointly infers the inducing inputs and the kernel hyperparameters by maximizing a lower bound of the true log marginal likelihood is introduced. A key property of this formulation is that the inducing inputs are defined to be variational parameters. On a different approach Williams & Seeger (2000) introduce a Nystrom approximation of the eigendecomposition of the Gram matrix across inducing variables (*i.e.*context pairs). Finally, the work of Tipping (1999) introduces the Relevance Vector Machine (RVM) which can be thought of as a Sparse version of the Support Vector Machine embedded within a Bayesian learning framework. Note that since the RVM is a finite linear model with Gaussian priors on the weights, it can be seen as a Gaussian Process.

An interesting direction of future work is to draw inspiration from Sparse GP approximations to formulate a learning objective that jointly learns the generative model in DPF as well as the distribution of context pairs in a hierarchical manner.

## G    LIMITATIONS AND FUTURE WORK

While DPF makes progress toward learning distributions over fields, there exist some limitations and opportunities for future work. The first limitation is that a vanilla implementation of the score network as a transformer becomes computationally prohibitive even at low resolutions, due to the quadratic cost of computing attention over context and query pairs. To alleviate this issue, we leverage the PerceiverIO (Jaegle et al., 2022) architecture that scales linearly with the number of query and context pairs. To further address this, one possible direction of future work is to explore other efficient transformer architectures (Zhai et al., 2022). In addition, similar to DDPM (Ho et al., 2020), DPF iterates over all time steps during sampling to produce a field during inference, which is slower

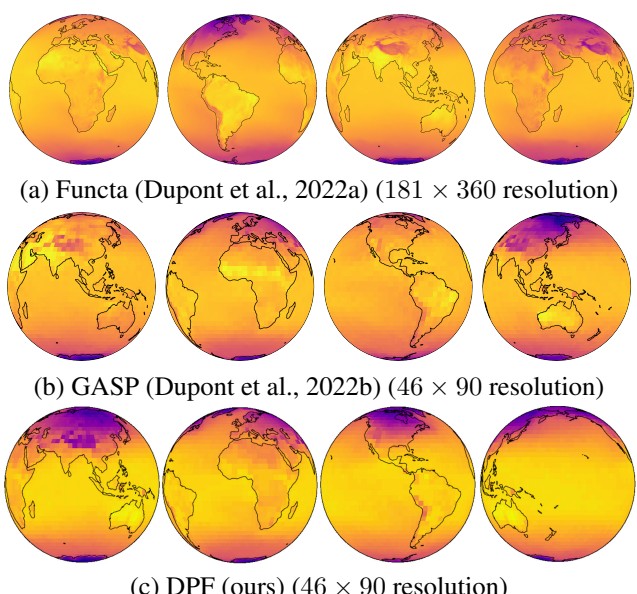

(a) Functa (Dupont et al., 2022a) ($181 \times 360$ resolution)

(b) GASP (Dupont et al., 2022b) ($46 \times 90$ resolution)

(c) DPF (ours) ($46 \times 90$ resolution)

Figure 15: **Qualitative comparison** of different implementations of the score field for fields defined on $\mathbb{S}^2$. (a) Functa (Dupont et al., 2022a). (b) GASP (Dupont et al., 2022b). (c) DPF (ours).

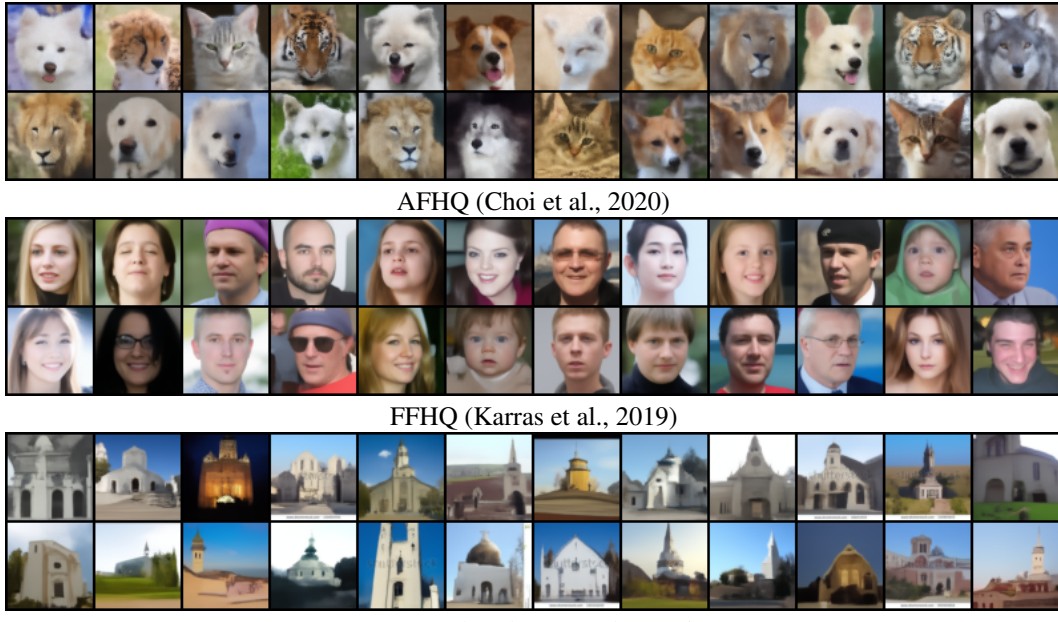

AFHQ (Choi et al., 2020)

FFHQ (Karras et al., 2019)

LSUN Church (Yu et al., 2015)

Figure 16: **Additional results on image datasets**. We show generated results using DPF on AHFQ (Choi et al., 2020), FFHQ (Karras et al., 2019) and LSUN church (Yu et al., 2015) data at $32 \times 32$ resolution. Face images are generated from DPF, *i.e.*, we do not directly reproduce face images from FFHQ (Karras et al., 2019).

compared to GANs. Existing work accelerates sampling (Song et al., 2021a) trading-off sample quality and diversity. We highlight that improved inference approaches like (Song et al., 2021a) are trivially integrated with DPF.

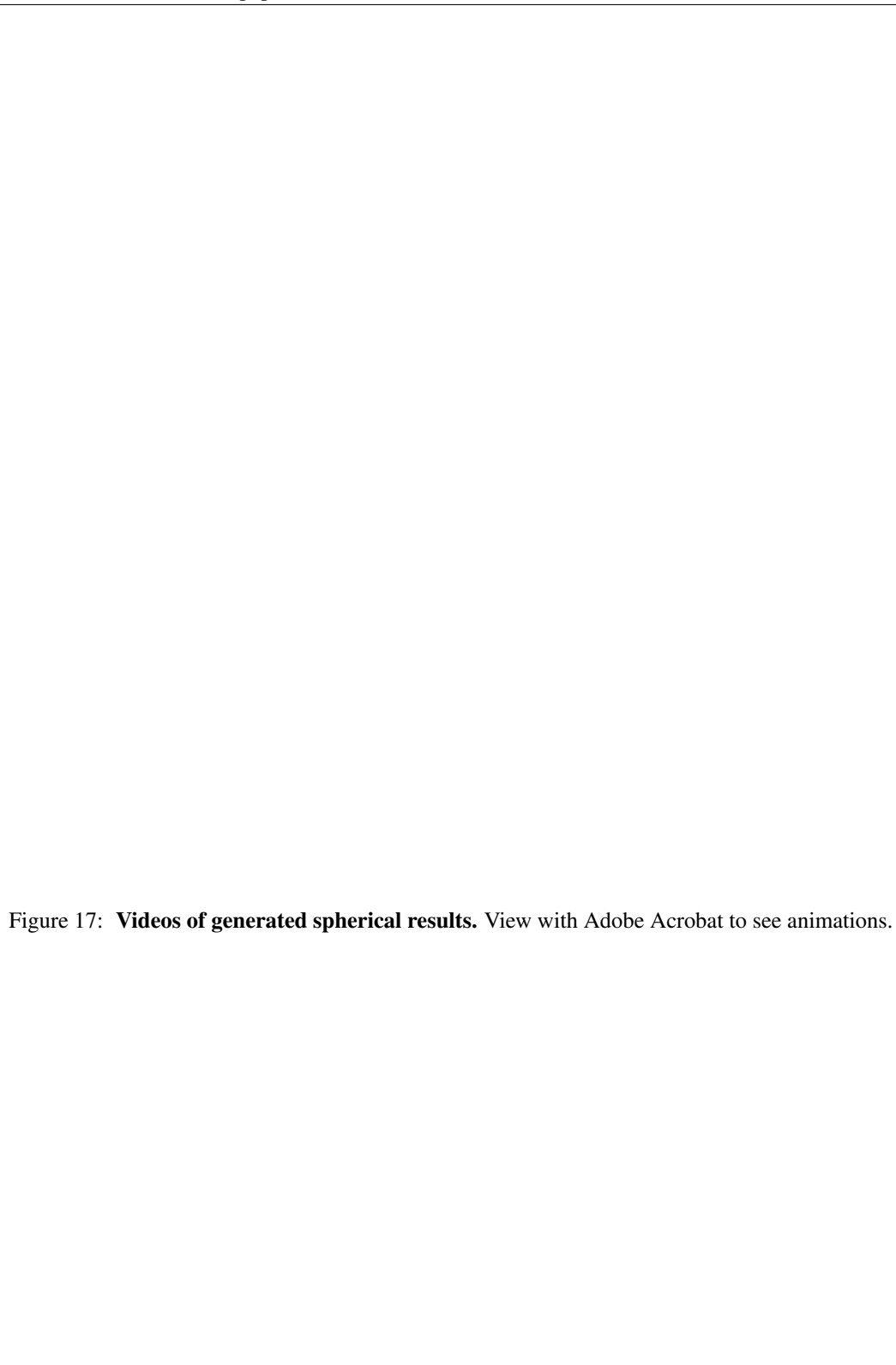

.

Figure 17: **Videos of generated spherical results.** View with Adobe Acrobat to see animations.

