# OpenReview forum: "Diffusion Probabilistic Fields"
_ICLR.cc/2023/Conference — ICLR 2023 poster_

### Official Review · Reviewer_6arD · 2022-10-15

**Confidence:** 4
**Correctness:** 4
**Technical Novelty And Significance:** 3
**Empirical Novelty And Significance:** 3
**Recommendation:** 8

**Clarity, Quality, Novelty And Reproducibility:**

**Clarity**

While most of the paper is generally clear and the figures are nice, the architecture of the score network and the treatment of context and query pairs could be further clarified.

**Quality**

The authors present an interesting method with a nice experimental evaluation and a good discussion of related work. As such I believe this is a quite high quality paper.

**Novelty**

While generative modeling of neural fields has been explored before, the authors provide a fairly novel method for learning diffusion models directly on neural fields (without an intermediate stage of mapping the fields to a latent space).

**Reproducibility**

The details and explanations provided in the paper seem to be sufficient for reproducing the results in the paper. In addition, the appendix contains detailed tables of hyperparameters for all experiments.

**Details Of Ethics Concerns:**

No ethical concerns.

**Strength And Weaknesses:**


**Strengths**

- The paper tackles an interesting and timely problem. With the recent surge of interest both in neural fields and diffusion models, I believe this paper will be of interest to the ICLR community
- The method is fairly novel. Previous works on learning distributions of neural fields have used a two stage approach (embedding neural fields into a latent vector and then learning a distribution over these) which has limitations discussed by the authors. Learning directly on coordinate and feature pairs has been considered in the context of GANs before with GASP, but the results in this paper are considerably stronger
- The experimental results are generally thorough and interesting. The model performs favorably compared to several recent baselines
- There is a good discussion of related work and the paper is quite well situated in the literature
- I like the ablation comparing each element of DDPM and the proposed model in the appendix. It's interesting to see what "makes the model work"
- It is nice to see how seamlessly the model works on spherical data

**Weaknesses**

- While most of the model is fairly well explained, it was not very clear to me exactly how context and query pairs were treated differently until reading the appendix (e.g. in Algorithm 1 and Figure 3, context and query points are treated exactly the same way). I think the paper would benefit from a clearer explanation of this as it is quite crucial to how the model works. It could for example be useful to have a diagram of the score network either in the main text or appendix to clarify this as it's currently not very clear
- There are no qualitative or quantitative comparisons for the spherical data experiments. As both functa and GASP consider generative modeling on spherical data it would be useful to have these as baselines to compare to
- The model appears to be computationally expensive which I believe is quite important and deserves to be discussed in more detail in the main text. The two stage process used by GEM and functa for example makes training a generative model fairly cheap, as the model is only fed a relatively small latent vector. Further, the authors only train on fairly small images (up to 64x64) and still require 8 A100 GPUs to train, which is quite heavy. While it is okay for the model to have this weakness, it should be more openly acknowledged in the main text
- The results do not match SOTA compared to specialized models on images for example. However, as learning generative models on neural fields is a fairly new problem, I don't believe it is very crucial for these models to outperform highly optimized convolutional models yet. This is therefore only a minor weakness in my opinion
- The proposed model is quite closely related to neural processes (also modeling distributions of fields and using a context/query pair setup). I think the paper would benefit from a more thorough discussion of neural processes in the related work. While I realise there are space constraints, I don't think the extended discussion of GANs is necessary, so this could be replaced with a discussion of neural processes
- There is some confusion about the naming of baselines. I believe a better naming convention would be functa for Dupont 2022a and GASP for Dupont 2022b. I'm not sure exactly why FDN was chosen as the name for functa. Further, it seems that for some reason in the GEM paper, the authors refer to GASP as FDN, whereas in this paper the authors refer to functa as FDN, leading to even more confusion (I am not sure where FDN comes from as it does not seem to be mentioned in either of Dupont 2022a/b). In addition, in the 3D experiments, the baseline is incorrectly referred to as FDN (Dupont 2022a), when it is actually GASP (Dupont 2022b). In short, I think it would be better to call Dupont 2022a functa (or something similar and more intuitive). In addition, the name of the baseline in the 3D geometry experiments is incorrect and should be changed to GASP (as it's referred to in the rest of the paper).

**Summary Of The Paper:**

This paper introduces a method for learning diffusion models over neural fields. More specifically, the authors perform diffuson directly on coordinate (such as pixel locations) and feature (such as RGB values) pairs, allowing the model to handle a wide variety of data that can be expressed as a neural field. The authors evaluate their model on small image datasets, 3D shapes and spherical images, obtaining favorable results compared to recent baselines.

**Summary Of The Review:**

Overall I think this is a fairly strong paper which will be of interest to the ICLR community. The model is timely and the experimental results are quite convincing. While there are some issues around clarity and a few other minor weaknesses, I still believe the strengths outweigh the weaknesses and so I recommend a weak accept.

---

> ### Author Response · Authors · 2022-11-15
> **Rebuttal comments**
>
> We thank the reviewer for their comments, which we address as follows.
>
> **Q: "a clearer explanation of how context and query pairs were treated differently. It could for example be useful to have a diagram of the scoring network either in the main text or appendix to clarify this as it's currently not very clear."**
>
> A: We thank the reviewer for this comment, we have included an explanatory figure in the updated version of the appendix showing how context and query pairs are treated in the score network (Fig. 11).
>
>
> **Q: "No qualitative or quantitative comparisons for the spherical data experiments."**
>
> A: In Sect. E of the updated version we provide a comparison with Functa and GASP on the ERA5 climate dataset. Following Functa we keep this comparison qualitative. In summary, DPF produces samples that are visually indistinguishable from GASP or Functa.
>
> DPF captures the same patterns as Functa and GASP, where the poles have cooler temperatures (eg. dark purple tones) and the equator and tropics have considerably high temperatures shown by the lighter yellow color. We believe an interesting direction of future work is to develop quantitative distributional metrics for signals over the sphere.
>
> **Q: "The model appears to be computationally expensive which I believe is quite important and deserves to be discussed in more detail in the main text. The two-stage process used by GEM and functa for example makes training a generative model fairly cheap, as the model is only fed a relatively small latent vector. Further, the authors only train on fairly small images (up to 64x64) and still require 8 A100 GPUs to train, which is quite heavy. While it is okay for the model to have this weakness, it should be more openly acknowledged in the main text."**
>
>
> A: We have added a short discussion of the computational cost of different approaches in the related work section. We agree with the reviewer that two-stage approaches make training the generative model in the second stage computationally cheaper than DPF.
>
> At a high level, two-stage approaches like Functa and GEM solve a compression task in the first stage and a probabilistic task in the second one. As a result, it is possible that two stage approaches will suffer when trained on datasets with a large number of samples, due to the fact compressing fields/samples into small fixed-length vectors in the first stage becomes more difficult as the number of samples increases. This is because the weights of the implicit model in the first stage need to be amortized across samples. With DPF, however, we were able to successfully train on large datasets like LSUN Churches (at 32x32 res, see Fig. 12) which contains 126227 samples. To the best of our knowledge, this is ~4x bigger than the biggest dataset used in GEM, Functa, or GASP.
>
> We believe a promising direction of future work is to couple DPF with efficient architectures like MLP mixers, which can enable training DPF on datasets containing a large number of samples/fields at high resolutions.
>
> **Q:"Didn’t match SOTA compared to specialized models."**
>
>
> A: We thank the reviewer for this comment. We believe that there are interesting future directions for bridging the gap between domain-specific and domain-agnostic architectures. In particular, it would be interesting to explore inductive biases in positional embeddings to encode domain knowledge (eg. symmetries), as well as explore efficient architectures like transformers or MLP-based architectures like MLP-mixers. Finally, we expect that when training generalist models on large amounts of data the inductive biases of domain-specific architectures will not be as important.
>
>
> **Q:"Replacing GAN discussion with neural processes in the related work."**
>
> A: Thanks for the useful suggestion. We added an extended discussion of the neural process literature in the updated manuscript and cut down the discussion on GANs.
>
>
> **Q: "Baseline naming. In addition, in the 3D experiments, the baseline is incorrectly referred to as FDN (Dupont 2022a), when it is actually GASP (Dupont 2022b)."**
>
>
> A: We thank the reviewer for this important observation, we followed the naming in GEM (Du 2021) for consistency but now realize it was conflicting with previous work. We have changed the baseline naming of (Dupont 2022a) to be Functa instead of FDN and GASP for (Dupont 2022b) to avoid the conflict naming in GEM (Du 2021). We have also updated the 3D geometry results in Tab. 3 to reflect that the baseline is GASP (Dupont 2022b) and not Functa (Dupont 2022a).

---

> > ### Comment · Reviewer_6arD · 2022-11-20
> > **Thank you for your response**
> >
> > Thank you for your detailed and thorough response. The changes made to the paper address all my concerns and have significantly improved the paper. I have therefore increased my score from 6 to 8. In agreement with the other reviewers, I believe this is a strong paper which will be of interest to the ICLR community and as such recommend acceptance.

---

> > > ### Author Response · Authors · 2022-11-29
> > > **Thank you**
> > >
> > > We thank the reviewer for their time and consideration towards our paper. We believe the recommendations of the reviewer were really helpful to highlight the quality of our submission.

---

### Official Review · Reviewer_E4qq · 2022-10-24

**Confidence:** 4
**Correctness:** 4
**Technical Novelty And Significance:** 3
**Empirical Novelty And Significance:** 3
**Recommendation:** 6

**Clarity, Quality, Novelty And Reproducibility:**

The paper is generally clear, and it has some novel ideas since it extends the diffusion model to neural field representation of data. It seems the contribution is a bit insufficient (see above comments). Paper seems to have sufficient information for reproducibility. I hope code will be made available upon acceptance.

**Strength And Weaknesses:**

The field of representing data as neural field has recently seen popularity. This work is new iin diffusion models in the sense that it is probably the first to work with field representation of data, which is a positive aspect of the paper. Also, this paper uses the transformer architecture as a score function, which allows it to propose a common architecture for all kinds of data. This is a good contribution and might encourage more use of transformer architecture in the future.

However, the technical contribution of the paper seems to be limited. Mainly using coordinate signal pair and having a transformer architecture that can process such a paired information appropriately is the key idea. While training, the authors use context and query pairs. I believe a bit of more analysis, perhaps theoretical, on how to choose these context and query pairs in better way could strengthen the paper.

**Summary Of The Paper:**

The paper presents a diffusion model for an explicit neural fields. They extend the regular diffusion model training and sampling, but instead of assuming an explicit domain for the signal, like 2D grid for images, their model is general enough to work for all kinds of domain through neural field representation. That is possible because the diffusion model is learnt over those fields, $f : \mathcal{M} \to \mathcal{Y}$. Another key feature of the paper is that same score model architecture applies to all kinds of data like images, 3D geometric data or data on $\mathbb{S}^2$. This is achieved by using transformer as the score function.

**Summary Of The Review:**

The paper is easy to follow and extend the diffusion model to the field representation of data. The use of transformer architecture allows the  common architecture for data of different types. However, the contribution seems to be insufficient. So, I am slightly on the negative side.

---

> ### Author Response · Authors · 2022-11-15
> **Rebuttal comments**
>
> We thank the reviewer for their comments, which we address as follows:
>
> **Q: "Limited technical contribution; mainly using coordinate signal pair and having a transformer architecture that can process such paired information appropriately//The use of transformer architecture allows the common architecture for data of different types."**
>
> A: The core contribution is DPF is formulating a single stage generative model for field representations. In contraposition with previous work like Functa (Dupont 2022a) and GEM (Du 2021), which require two independent training stages: first, compressing fields into a fixed length latent vectors and then learning a distribution over these (which has limitations that we discuss at length in the intro section). The single stage formulation in DPF is indeed achieved by directly modeling field data in its raw form (as coordinate-signal pairs). We empirically show that this single stage formulation in DPF outperforms two stage approaches like Functa and GEM as well as adversarial field methods like GASP (Dupont 2022a). An additional note is that we don’t require to design modality specific decoders for training the first stage, which are oftentimes leveraged by two stage approaches.
>
> Furthermore, the single stage formulation in PDF is not constrained to be implemented with  transformer architecture. To support this claim, in the new version of the manuscript Sect. D we have included an ablation where we experiment with different implementations of the score network: a PerceiverIO, a vanilla transformer encoder, and an MLP-mixer. Our results show that the performance and generalizability of DPF are not a result of using a PerceiverIO architecture and that different architectures can be effectively used in the score field network.
>
>
> **Q: "More analysis, perhaps theoretical, on how to choose these context and query pairs in a better way could strengthen the paper"**
>
> A: We thank the reviewer for this comment. We agree that a deeper analysis of the problem of selecting context pairs is interesting and we have included a discussion on this in Sect. F.
> As a summary, sampling query pairs depend on the resolution at which the field needs to be sampled, and this is problem-dependent (eg. to generate a 32x32 image as a field, one needs to sample 1024 query pairs densely).
> On the other hand, the problem of selecting context pairs can be mapped to the problem of selecting inducing variables in Sparse Gaussian Processes, for which there exists an extensive body of work. In Sect. F we provide an overview of the main approaches to the problem of choosing inducing variables.
> We believe that an extended analysis of this issue is an interesting avenue of future work that can impact both DPF and Gaussian Processes.

---

> > ### Comment · Reviewer_E4qq · 2022-11-21
> > **Gaussian Process argument is new**
> >
> > Thanks for the reply. Yes, I agree that the framework is general and applicable to different data types.
> >
> > The sparseGP argument seems totally new. Many of the works in sparseGP frameworks including the ones that authors have mentioned were developed to resolve the computational complexity problem in GP and they enjoy interesting connection of matrix factorization and variational interpretation to achieve faster computation fo GP posterior. While I agree that some insights on point selection from there may be relevant, it is hard for me to understand how GP is related to the architecture proposed in this work.
> >
> > I agree that the work has some merit, especially when compared to the existing neural field methods. So, I will increase my score.

---

> > > ### Author Response · Authors · 2022-11-29
> > > **Thank you**
> > >
> > > We thank the reviewer for taking the time in the discussions and for providing feedback. We are happy to address any remaining concerns if needed before the final assessment.

---

### Official Review · Reviewer_NtLL · 2022-10-25

**Confidence:** 3
**Correctness:** 4
**Technical Novelty And Significance:** 3
**Empirical Novelty And Significance:** 3
**Recommendation:** 8

**Clarity, Quality, Novelty And Reproducibility:**

The work was mostly clear in my opinion, but lacked details on their main conceptual contributions, in my opinion. Some questions that lingered after a reading:
- How does the diffusion model operate differently than the classical DDPM, given the need for some sort of interplay between the context and query pairs? If one reads the Ho et al. paper, there is significantly more detail outlining the exact model and motivations behind the training and sampling procedures. Mostly, I'd like more detail on Equation (6) and how the dependency between context and query pairs is reflected in the loss terms and resulting algorithms.
- In line 7 of Algorithm 1, should $\epsilon_Q$ be $\epsilon_q$?
- What is the structure of the PerceiverIO, and is it necessary for good performance of the approach, or could another architecture have been used? Some of the explanations from the appendices should be part of the main text, I believe.

**Strength And Weaknesses:**

Strengths:
- The approach is the first diffusion-based method that does not use latent vectors to parameterize fields, and hence does not require a step to train for the distributions over latent vectors.
- The method seems to perform fairly well compared to other domain-agnostic models.

Weaknesses:
- The clarity of their model explanation with context and query pairs could be improved, I feel.
- The PerceiverIO framework could also use some explanation, even if brief, given that it's a key part of their method.

**Summary Of The Paper:**

The paper adapts a generative diffusion approach to neural fields, allowing simple adaptation to different data domains. A diffusion process is modeled on both context and query pairs and training is done with a score network that incorporates both sets of pairs. This framework is one of a few domain-agnostic approaches, and is the only diffusion-based method which does not require a latent vector to specify the field.

A PerceiverIO architecture is used for the score network, and experiments are done with image data, volumetric data, and spherical data. The method performs well when compared to other domain-agnostic approaches, on both quantitative and qualitative comparisons.

**Summary Of The Review:**

The work provides an interesting diffusion-based approach for generating neural fields. The clarity could use some work, but the empirical results are compelling, and I feel it is worth publication.

---

> ### Author Response · Authors · 2022-11-15
> **Rebuttal comments**
>
> We thank the reviewer for their comments, which we address in the following.
>
> **Q: "The clarity of their model explanation with context and query pairs could be improved, I feel."**
>
> A: In the new version of the manuscript we have included a figure (Fig. 11) with a detailed explanation of how context and query pairs are used in the score field network.
>
>
> **Q: "The PerceiverIO framework could also use some explanation, even if brief, given that it's a key part of their method."**
>
> A: We thank the reviewer for this comment. In the new version of the manuscript, we have included a brief explanation of the PerceiverIO framework in the method section and referenced an extended explanation, together with a figure in the appendix (Fig. 11).
>
> **Q: "How does the diffusion model operate differently than the classical DDPM, given the need for some sort of interplay between the context and query pairs? If one reads the Ho et al. paper, there is significantly more detail outlining the exact model and motivations behind the training and sampling procedures. Mostly, I'd like more detail on Equation (6) and how the dependency between context and query pairs is reflected in the loss terms and resulting algorithms."**
>
> A: We have provided a brief explanation of how context and query pairs interact  (Eq.6 is now inline due to space constraints of the main text) in the method section. In addition, we have also referenced an extended explanation in Fig. 11 in the updated appendix using a PerceverIO as a particular case for the implementation of the score network.
>
> Our original motivation for not explicitly describing these interactions in the main text is that these are architecture-dependent and DPF is a general formulation that allows for different implementations of the score field (as we show in the updated version of the manuscript in Sect. D).
>
> **Q: "In line 7 of Algorithm 1, should ϵQ be ϵq?"**
>
> A: We thank the reviewer for pointing out this typo, which has been corrected in the new version of the manuscript.
>
> **Q: "What is the structure of the PerceiverIO, and is it necessary for good performance of the approach, or could another architecture have been used? Some of the explanations from the appendices should be part of the main text, I believe."**
>
> A: The formulation of DPF is agnostic to the implementation of the score network, and there’s a large design space for such architectures. In particular, all architectures that can process data in the form of sets, like transformers or MLPs, are viable candidates. We have included a brief explanation of this observation in the method section of the new version of the submission.
>
> In addition, in the new version of the manuscript Sect. D we have included an ablation where we experiment with different implementations of the score network: a PerceiverIO, a vanilla transformer encoder and an MLP-mixer. Our results show that the performance and generalizability of DPF is not a result of using a PerceiverIO architecture and that different architectures can be effectively used in the score field network.

---

> ### Author Response · Authors · 2022-11-29
> **Addressing remaining concerns**
>
> We again thank the reviewer for their time and for their reviews of our submission. We hope our rebuttal successfully addresses the reviewers suggestions. We are happy to further address any remaining concerns not covered in the rebuttal that the reviewer may have before making their final assessment.

---

### Official Review · Reviewer_joBr · 2022-10-26

**Confidence:** 3
**Correctness:** 3
**Technical Novelty And Significance:** 2
**Empirical Novelty And Significance:** 3
**Recommendation:** 6

**Clarity, Quality, Novelty And Reproducibility:**

Main implementation details are in the Appendix B. Clarifying that the field is implemented as a transformer early on would benefit the reader understanding of the approach.

**Strength And Weaknesses:**

Strenghts

- The method is elegant in principle and domain agnostic yet obtains good performance

Weaknesses

- It is not clear how the gradient at step 7 is computed. $epsilon_Q$ is not defined.


**Summary Of The Paper:**

This paper presents an extension of Diffusion Probabilistic Models using Fields to model data.
The basic idea is to model data as functions (fields) e.g. images are functions from $R^2$ to $R^3$, a map between pixel coordinates and color pixel value.
This approach is motivated by the need to unify the generative model architecture to cover different modality, thus avoiding the need to construct specific architecture for score functions $\epsilon_\theta$ which are the denoisers used in the generation process.
Fields are parameterized using pairs of coordinates in source and target domains and evaluated in a similar manner using pairs of query pairs of coordinates.

Results are compelling considering that the Field Model is domain agnostic.

In the end the method is using a Transformer (Perceiver) to remap coordinates (implementing the field) (Appendix B)


**Summary Of The Review:**

Overall the paper is interesting and results are compelling. It has to be noted that the approach claims to be general with respect to the data in input but I feel that the main aspect is that a very flexible and powerful architecture (PerceiverIO) is used as a score function thus enabling this feature. The whole modelling of data as fields seems less relevant than this implementation detail.

---

> ### Author Response · Authors · 2022-11-15
> **Rebuttal comments**
>
> We thank the reviewer for their comments, which we address in the following.
>
> **Q: "It is not clear how the gradient at step 7 is computed. epsilon_Q is not defined."**
> A: We thank the reviewer for mentioning this typo. The variable should be $\epsilon_q$ instead of $\epsilon_Q$. We have fixed this typo in the new version of the manuscript (Alg. 1). In terms of the gradient, it is computed wrt the score field network parameters $\theta$ using the denoising objective.
>
> **Q: “Clarifying that the field is implemented as a transformer early on would benefit the reader's understanding of the approach.”**
>  A: We thank the reviewer for this suggestion. In the updated version of the draft, we have included a small paragraph in the method section (page 5) describing different implementation choices and our motivation for choosing a transformer.
>
> **Q: “It has to be noted that the approach claims to be general with respect to the data in input but I feel that the main aspect is that a very flexible and powerful architecture (PerceiverIO) is used as a score function thus enabling this feature.”**
> A: We thank the reviewer for this comment, which we also believe is important to clarify. Note  that the formulation of DPF is agnostic to the choice of implementation in the score field network. In particular, all architectures that can deal with data in the form of sets are viable candidates for the score field model. The generality with respect to data input is not due to a particular implementation choice (eg. a PerceiverIO) but rather the formulation of DPF.
>
> To support this claim, in the new version of the manuscript Sect. D we have included an ablation where we experiment with different implementations of the score network: a PerceiverIO, a vanilla transformer encoder, and an MLP-mixer. Our results show that the generality of DPF is not a result of using a PerceiverIO architecture and that different architectures can be effectively used in the score field model.

---

> ### Author Response · Authors · 2022-11-29
> **Addressing remaining concerns**
>
> We again thank the reviewer for their time and for their reviews of our submission. We hope our rebuttal successfully addresses the reviewers suggestions. We are happy to further address any remaining concerns not covered in the rebuttal that the reviewer may have before making their final assessment.

---

### Author Response · Authors · 2022-11-15
**To all reviewers**

We thank all reviewers for their thoughtful feedback. We modified the manuscript based on their suggestions. We also replied to all the reviewers' questions in detail and included corresponding discussions and experiments in the revised submission, which we hope reviewers take into consideration in their final assessment. We have highlighted the updated content in red color. We thank reviewers for their time.
In summary, reviewers have highlighted the following aspects of our submission:

- NtLL:” The work provides an interesting diffusion-based approach for generating neural fields”. “The method seems to perform fairly well compared to other domain-agnostic models”
- joBr: “The method is elegant in principle and domain agnostic yet obtains good performance“. “Overall the paper is interesting and results are compelling”
- E4qq: ”This work is new in diffusion models in the sense that it is probably the first to work with field representation of data, which is a positive aspect of the paper.”
- 6arD: “I believe this is a quite high-quality paper”.“Overall I think this is a fairly strong paper that will be of interest to the ICLR community.”

---

### Decision · Program_Chairs · 2023-01-20

**Decision:**

Accept: poster

**Justification For Why Not Higher Score:**

The proposed domain-agnostic approach does not meet SOTA performance of existing domain-specific approaches.

**Justification For Why Not Lower Score:**

This is an interesting and timely work, worth sharing to the community.

**Metareview: Summary, Strengths And Weaknesses:**

The paper proposes Diffusion Probabilistic Fields (DPF), a diffusion model learned over explicit neural fields. Specifically, it parameterizes fields explicitly by performing a diffusion process on coordinate and signal pairs, enabling the model to handle different modalities. The method shows compelling quantitative/qualitative experimental results compared to other domain-agnostic models.

The authors were quite responsive in addressing the reviewers' concerns in the author-reviewer discussion phase, and at the end, all reviewers unanimously supported the acceptance. AC also agrees with that, and given the recent advances in neural fields and current interest in diffusion models, the paper addresses an under-explored but timely problem and presents a promising direction to guide future research on this topic. Overall, AC is happy to recommend the acceptance.

**Note From Pc:**

if the above contains the word "oral" or "spotlight" please see: "oral" presentation means -> notable-top-5% and "spotlight" means -> notable-top-25%. As stated in our emails, we are disassociating presentation type from AC recommendations